# Energy landscape reshaped by strain-specific mutations underlies epistasis in NS1 evolution of influenza A virus

Iktae Kim [1], Alyssa Dubrow[1], Bryan Zuniga [1], Baoyu Zhao[1], Noah Sherer[1], Abhishek Bastiray [1], Pingwei Li[1] & Jae-Hyun Cho [1] ✉

Elucidating how individual mutations affect the protein energy landscape is crucial for understanding how proteins evolve. However, predicting mutational effects remains challenging because of epistasis—the nonadditive interactions between mutations. Here, we investigate the biophysical mechanism of strain-specific epistasis in the nonstructural protein 1 (NS1) of influenza A viruses (IAVs). We integrate structural, kinetic, thermodynamic, and conformational dynamics analyses of four NS1s of influenza strains that emerged between 1918 and 2004. Although functionally near-neutral, strain-specific NS1 mutations exhibit long-range epistatic interactions with residues at the p85β-binding interface. We reveal that strain-specific mutations reshaped the NS1 energy landscape during evolution. Using NMR spin dynamics, we find that the strain-specific mutations altered the conformational dynamics of the hidden network of tightly packed residues, underlying the evolution of long-range epistasis. This work shows how near-neutral mutations silently alter the biophysical energy landscapes, resulting in diverse background effects during molecular evolution.

Addressing the relationship between mutations and their effects on the functional landscape can provide a detailed evolutionary trajectory of a protein[1]. However, nonadditive interactions between mutations make comprehensive mapping of this sequence-function space extremely difficult to achieve, if not impossible[2].

For example, the effects of two single mutations at sites a and b, denoted as $\Delta G_a$ and $\Delta G_b$, respectively, can cause the double mutant of the two sites to show either additive ($\Delta\Delta G_{a,b} = \Delta G_a + \Delta G_b$) or non-additive behaviors ($\Delta\Delta G_{a,b} \neq \Delta G_a + \Delta G_b$)[3]. Epistasis refers to the non-additive effect between two mutations. Various patterns of epistatic interactions are possible, and the same mutation can result in different outcomes in different backgrounds[3,4]. Epistasis also allows neutral or near-neutral mutations to have an evolutionary impact by diversifying genetic backgrounds, which might serve as a pre-adapted platform in the new environment[5–7]. Alternatively, these mutations may have permissive roles in which subsequent mutations would otherwise be deleterious[8,9].

Although many studies on epistasis have focused on the mutational effects on the functional landscape, mapping the mutational effect on the energy landscape is critical for understanding the mechanistic basis of epistasis[10,11]. In particular, because neutral mutations are functionally silent on their own, deep mechanistic studies on how they alter biophysical traits of a protein are essential[6,8].

From the physical perspective, epistasis can be classified as a short- or long-range mutational effect. While a short-range epistatic interaction between directly contacting residues is structurally intuitive, the mechanistic basis of long-range epistatic interactions between spatially distant residues requires further refinement, although some interesting mechanisms are available[12–15]. Moreover, epistatic effects of neutral mutations often occur through long-range interactions[8,16]. Thus, the biophysical foundation of long-range epistasis can help understand how neutral mutations contribute to protein evolution[9].

In this study, we trace evolutionary changes in the in vitro functional and biophysical traits of nonstructural protein 1 (NS1) of

[1]Department of Biochemistry and Biophysics, Texas A&M University, College Station, TX 77843, USA. ✉e-mail: jaehyun.cho@agnet.tamu.edu

influenza A viruses (IAV). To this end, we selected four NS1s of human IAV strains that emerged between 1918 and 2004: 1918 H1N1, Puerto Rico 8 (PR8) H1N1, Udorn (Ud) H3N2, and Vietnam (VN) H5N1.

NS1 of an influenza virus is a multifunctional virulence factor that antagonizes the host's innate immune response and is associated with many strain-specific functions in viral replication and host range[17,18]. Moreover, NS1 is one of the most frequently mutated[19] and mutation-tolerant proteins[20] in the IAV genome. Multiple strain-specific mutations in NS1 were identified as adaptive mutations with respect to viral virulence and replication[21,22]; however, many remain uncharacterized, mainly because of the lack of functional phenotypes. Especially, many mutations occur in the effector domain (ED) of NS1 that interacts with a number of host factors, including retinoic acid-inducible gene 1 (RIG-I), tripartite motif containing 25 (TRIM25), and phosphoinositide 3-kinase (PI3K), during the infection cycle[23-26].

Outlining the present study, we first present that strain-specific mutations in NS1 EDs have near-neutral effects on the association kinetics to the p85β subunit of PI3K. We subsequently find that near-neutral, strain-specific mutations have long-range epistatic interactions with the residues on the p85β-binding interface of NS1. Using isothermal titration calorimetry (ITC), we then find that strain-specific mutations altered the thermodynamic energy landscape of the NS1:p85β interaction. To further address how strain-specific mutations altered the energy landscape of NS1s and enabled long-range epistasis during evolution, we conducted extensive biophysical analyses of NS1s. Structural analyses revealed conformational variations occurred during NS1 evolution. Especially, using NMR spin dynamics of backbones and side-chains, we reveal how strain-specific mutations influence long-range epistasis through dynamic reorganization of packing

interactions in the hydrophobic core. Consequently, our study provides a high-resolution mechanism by which neutral mutations contribute to protein evolution by reshaping the energy landscape, despite the absence of functional changes.

## Results

### Strain-specific epistasis in NS1

The four NS1 EDs (referred to as NS1s) employed in this study contain 3–14 % of residues that are strain-specifically mutated in the protein sequence with respect to the 1918 NS1 (Supplementary Fig. 1a). Many of the mutations are conservative between similar types of amino acids located at buried or partially buried sites in the NS1 structure; the average relative solvent accessible surface area of mutated residues was 32.2% (Fig. 1a, b and Supplementary Fig. 1b).

To study the functional effect of strain-specific mutations, we first confirmed that all four NS1s bind to the full-length PI3K using co-immunoprecipitation (Supplementary Fig. 2a). Next, we characterized the interaction between NS1 and p85β using BLI (Biolayer interferometry). Compared to the 1918 NS1, other NS1s dissociated from p85β with a smaller dissociation rate constant, resulting in a moderately higher binding affinity to p85β (Fig. 1c and Supplementary Figs. 2b–e).

However, there was no difference in the binding among the later strains of 1918 IAV; namely, PR8, Ud, and VN NS1s shared virtually identical binding ($k_{on}$) and unbinding ($k_{off}$) rate constants (Fig. 1c). Notably, the $k_{on}$ value was highly similar in all four NS1s, including 1918 NS1, showing that strain-specific mutations across the NS1s have a near-neutral effect on the p85β-binding rate constant. These results also suggest that $k_{on}$ is a biophysical constraint of NS1 evolution[27,28].

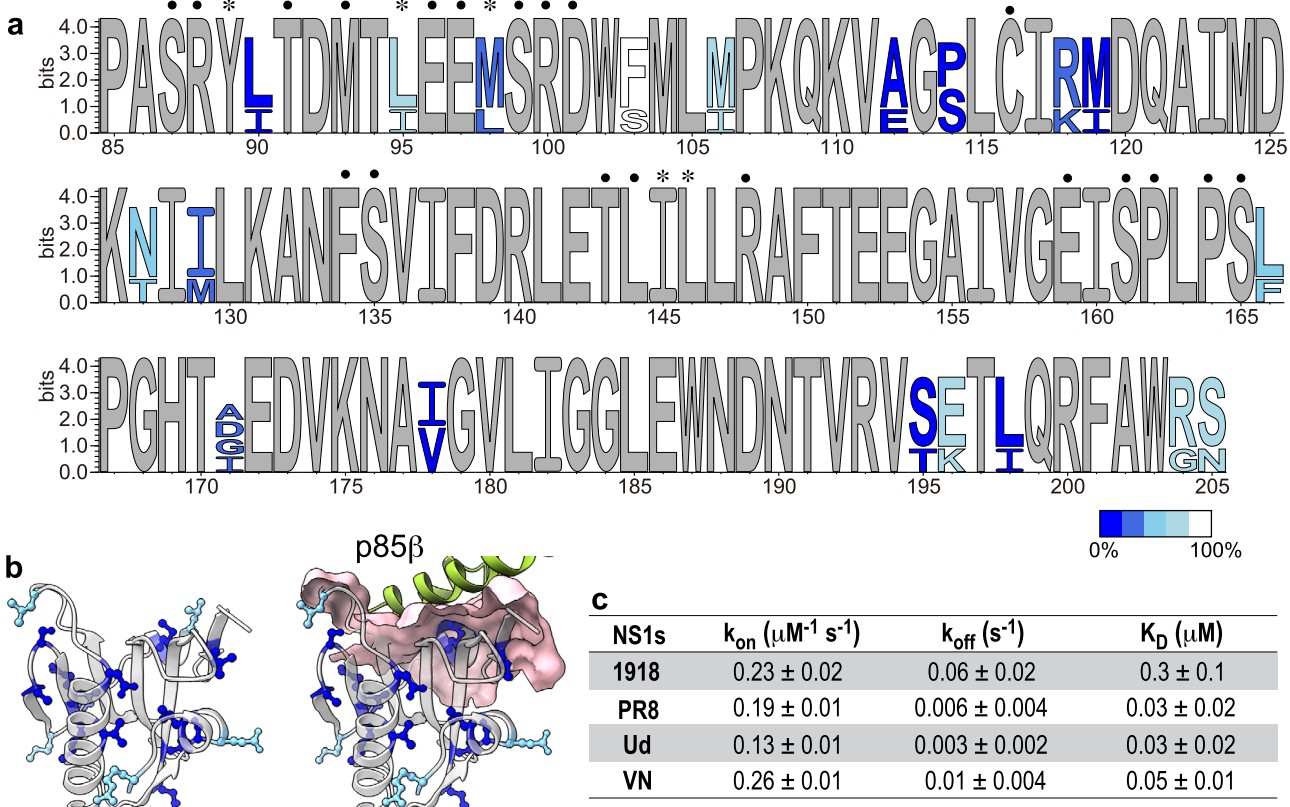

**Fig. 1 | Strain-specific mutations of NS1s. a** The sequence logo of four NS1s. Completely conserved residues are in gray. Mutated residues are color coded according to the relative solvent accessible surface area. Asterisks and black filled circles represent the core and rim interface residues, respectively. **b** (left panel) Positions of strain-specific mutations are shown as ball-and-sticks. Partially or completely buried residues are shown in blue. Solvent exposed residues are shown in light blue. (right panel) The p85β-binding interface is shown in surface model. The bound p85β is shown green. **c** BLI-based kinetic parameters for binding between NS1s and p85β. The uncertainties correspond to the standard deviation of three repeated measurements.

| NS1s | $k_{on}$ (μM$^{-1}$ s$^{-1}$) | $k_{off}$ (s$^{-1}$) | $K_D$ (μM) |
|------|------|------|------|
| 1918 | 0.23 ± 0.02 | 0.06 ± 0.02 | 0.3 ± 0.1 |
| PR8 | 0.19 ± 0.01 | 0.006 ± 0.004 | 0.03 ± 0.02 |
| Ud | 0.13 ± 0.01 | 0.003 ± 0.002 | 0.03 ± 0.02 |
| VN | 0.26 ± 0.01 | 0.01 ± 0.004 | 0.05 ± 0.01 |

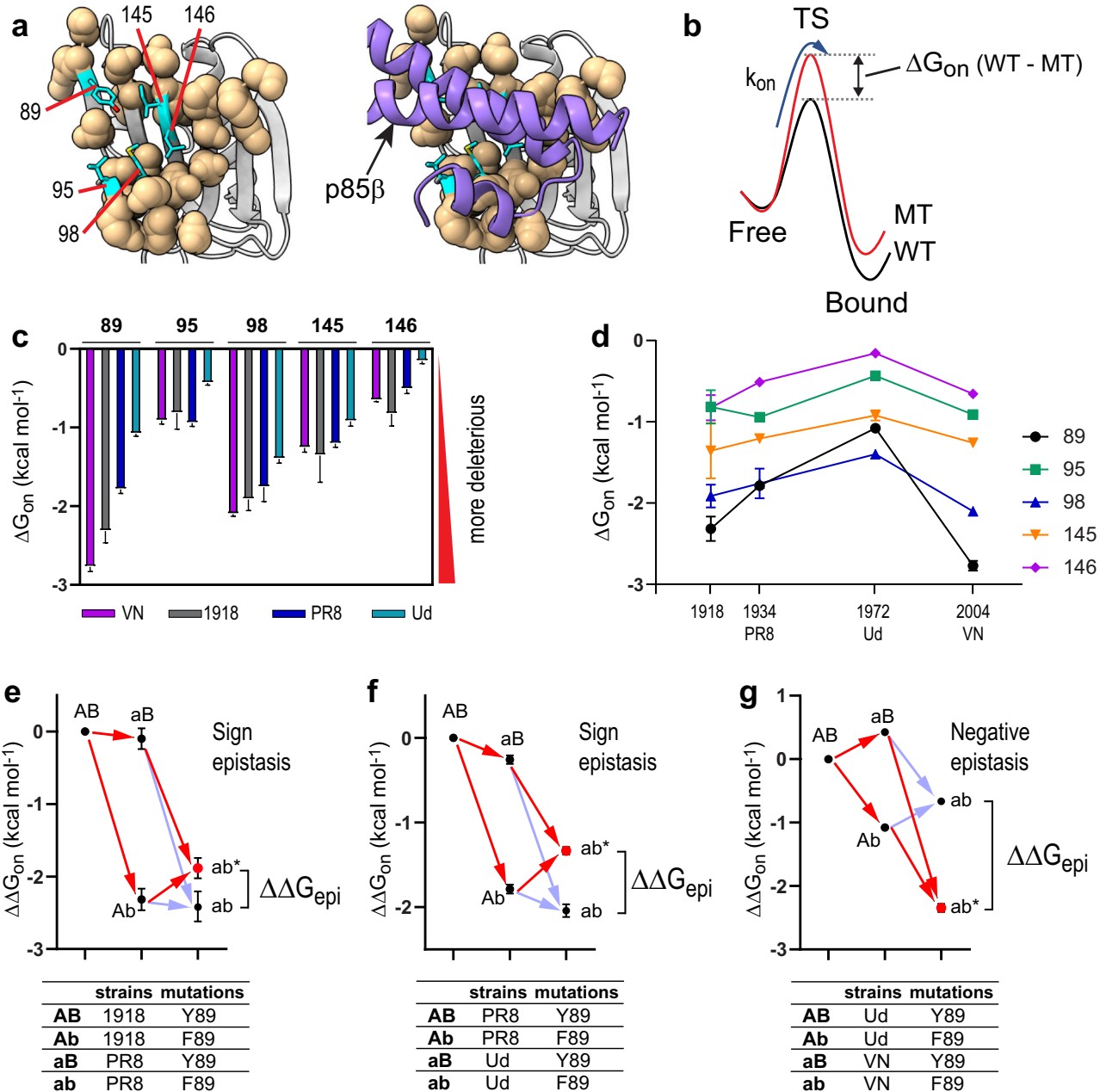

**Fig. 2 | Strain-specific epistasis in NS1s. a** The core and rim interface residues are shown as cyan sticks and wheat spheres, respectively. **b** Definition of $\Delta G_{on}$ (WT−MT) in the binding reaction coordinate between NS1 and p85β. Negative $\Delta G_{on}$ (WT−MT) indicates deleterious mutational effects on association kinetics ($k_{on}$). **c** Strain-specific $\Delta G_{on}$ of core interface residues. **d** Evolution of $\Delta G_{on}$ of core interface residues as a function of years of the occurrence of IAV strains. **e**–**g** Epistatic interactions ($\Delta\Delta G_{epi}$) between strain-specific mutations (A → a) and mutations of individual core interface residues (B → b). ab* corresponds to the experimentally measured effects of double mutations, while ab corresponds to the calculated additive (non-epistatic) effects. Red and faint blue arrows correspond to the experimentally measured and calculated additive effects of mutations, respectively. The difference between ab* and ab defines the pattern of epistasis. Epistatic interactions of Y89F mutation between **e** 1918 and PR8, **f** PR8 and Ud, and **g** Ud and VN NS1 backgrounds. **c**–**g** All bars or filled symbols represent $\Delta G_{on}$ or $\Delta\Delta G_{on}$ values calculated using mean $k_{on}$ values. Error bars represent the propagated standard deviation of three repeated measurements of $k_{on}$ values. Source data are provided as a Source data file.

We further tested whether the energetic contribution of the individual p85β-binding residues to $k_{on}$ is conserved across different NS1 backgrounds (Supplementary Fig. 3). The p85β-binding interface on NS1 consists of 20 residues based on the crystal structure of the complex[23,29], while our previous study identified residues 89, 95, 98, 145, and 146 of NS1 as the major players determining the p85β-binding rate constant[29]; thus, these residues are referred to as core interface residues (Fig. 2a).

Using Ala-scanning mutagenesis[30], we measured the extent to which individual core interface residues contribute to the p85β-

binding rate constant in the four NS1 backgrounds. For Y89, however, we incorporated F89 instead of A89 because the Y89A mutation reduced binding too much to quantify the effect. We first calculated $\Delta G_{on}$ (WT−MT) of the individual mutants with respect to their wild-type constructs (Fig. 2b); a larger negative number means a larger deleterious effect of a mutation. Notably, individual core interface residues have substantially different binding energetics in different NS1 backgrounds (Fig. 2c). Next, to follow the evolutionary trajectory of the binding energetics at the residue level, we plotted the $\Delta G_{on}$ of the core interface residues along with the year of occurrence of IAV

strains (Fig. 2d). Overall, the energetic contribution of core residues decreased from 1918 to 1972 (Ud) and then increased in 2004 (VN).

These background-dependent mutational effects suggest that the core binding residues are energetically coupled with strain-specific mutations of NS1, i.e., epistatic interactions. So, we measured the epistatic interactions using a thermodynamic cycle analysis (Fig. 2e–g and Supplementary Fig. 4). Note that we considered a different NS1 background as a single mutant, although multiple mutations are involved in different NS1s. Thus, energetic coupling (i.e., epistasis) in our study represents the interaction between mutations of individual core interface residues and the collection of strain-specific residues. Following the standard notation for epistasis[4], here, we define that additive and negative epistasis corresponds to less (i.e., larger $k_{on}$ values) and more (i.e., smaller $k_{on}$ values) deleterious effects, respectively, by a double mutation than the sum of two single mutational effects. Sign epistasis means that the effect of a double mutation is beneficial with respect to one type of single mutation background and deleterious with respect to another single mutation background.

Indeed, we observed various types of epistasis, including additive, negative, and sign epistasis Supplementary Fig. 4). Moreover, the same residue exhibited different types of epistasis depending on the choice of NS1 background. For example, Y89F showed sign epistasis between the 1918 and PR8 NS1 backgrounds but negative epistasis between Ud and VN NS1 backgrounds (Fig. 2e–g).

Then, to examine the evolutionary trajectory of epistasis, we calculated how the epistatic interaction evolved in different NS1s relative to the 1918 NS1 (Supplementary Fig. 5). This trajectory is equivalent to cumulative epistatic interactions starting in 1918. Overall, core interface residues showed positive epistatic interactions with strain-specific mutations between 1918 and 1972, indicating that the binding interface mutations have less deleterious effects on $k_{on}$ values of NS1s of later IAV strains. This trend reversed between 1972 and 2004, with large negative epistasis. As a result, overall epistatic interactions seemed to fluctuate around null epistasis. This outcome indicates that strain-specific mutations have near-neutral effects on the overall association kinetics, while they altered the energetic contribution of individual residues during NS1 evolution. Structurally, however, the majority of strain-specific mutations are located remotely from the p85β-binding interface, indicating long-range epistasis (Fig. 1b).

### Strain-specific mutations reshaped the energy landscape of NS1

Long-range epistasis—energetic coupling between distant residues—suggests that strain-specific mutations altered the protein energy landscape[3,4]. The protein energy landscape can be probed by examining biophysical properties such as binding thermodynamics[31]. Thus, we characterized the thermodynamic signature for the binding of NS1s to p85β using isothermal titration calorimetry (ITC). Here, changes in binding thermodynamic signature across NS1s represent the evolution of the binding energy landscape accompanying changes in enthalpic (ΔH) and entropic (TΔS) properties.

Overall, binding was driven by favorable enthalpy and entropy changes (Fig. 3a and b). However, Ud NS1 showed a striking difference from other NS1s in that the binding was not accompanied by measurable heat; that is, the binding was driven exclusively by a favorable entropy change (Fig. 3a, b). Although minor compared to Ud NS1, other NS1s also showed sizable variations in thermodynamic signature, indicating strain-specific variations in the binding energy landscape.

This result is surprising because the protein sequence of the p85β-binding interface is almost completely conserved across the NS1s. The only difference is that Ud NS1 contains mutations at two of the 20 interface residues compared to the other three NS1s. However, these mutations are isosteric: L to I and M to L for residues 95 and 98, respectively (Fig. 3c, d). Moreover, grafting the Ud sequence to 1918 NS1 did not affect its binding to p85β[23].

To further confirm that the two mutations are not responsible for the drastic change in the thermodynamic signature, we incorporated I95L/L98M mutations into Ud NS1; i.e., this mutant contains the same sequence in the binding interface as other NS1s. ITC data showed that the binding of Ud-I95L/L98M to p85β is not accompanied by heat as in wild-type Ud NS1 (Fig. 3a, b). We also tested the effect of grafting the Ud sequence to the PR8 background (i.e., PR8 L95I/M98L) and found no significant change in the thermodynamic signature for binding to p85β compared to that of wild-type PR8 NS1 (Fig. 3a, b). These results demonstrated that the direct binding interface is not responsible for the variation in the thermodynamic signature for binding. Instead, our results suggested that strain-specific mutations reshaped the energy landscape of NS1 via long-range interactions, resulting in strain-dependent thermodynamic signatures for binding to p85β. However, its underlying mechanism remains to be determined. In particular, future studies on the NS1-p85β complex are warranted to reveal the mechanism.

These results are reminiscent of the recent finding that proteins explore diverse thermodynamic mechanisms by neutral or selective mutations during adaption to high temperatures, referred to as thermodynamic systems drift[32]. Thus, our results support the notion that proteins can acquire the same functional phenotype using completely different biophysical mechanisms[32–34].

### NS1s have the same structures in the complexes with p85β

To reveal the mechanistic bases of long-range epistasis and how strain-specific mutations reshaped the energy landscape of NS1 during evolution, we first examined the structure of the NS1:p85β complexes. Noticeable changes in the complex structures could be a plausible explanation for the strain-specific epistasis and variation in the thermodynamic signature. The structure of the NS1:p85β complex was available for 1918 and PR8 NS1s[23,35], and we determined a new crystal structure of VN NS1 bound to p85β in this study (Supplementary Table 1).

Interestingly, the conformations of NS1s in the complexes were almost identical, with RMSD < 0.5 Å (Fig. 4a, b). Although the complex structure of Ud NS1 is not available, it is reasonable to assume that it has a structure similar to other NS1s, considering the sequence identity in the binding interface. This result indicated that the complex structure is not responsible for strain-specific mutational effects on NS1.

Given the structural invariance of the complexes, we hypothesized that the structure and dynamics of NS1 in the unbound state are most likely responsible for the strain-specific epistasis and thermodynamic signature for binding. The following observations further support this hypothesis. The observed epistasis represents the change in the free-energy difference between the unbound and transition states (Fig. 2b), and the ITC result corresponds to the free-energy difference between the unbound and bound states. Therefore, both results are directly related to the structure and dynamics of free NS1.

### Conformational energy landscape of free NS1

To study the evolutionary influence on the conformation of free NS1, we first compared all the crystal structures of free NS1s (n = 16) of the four IAV strains available in PDB. The structural alignment showed substantial conformational heterogeneity among free NS1s (Fig. 4a, c). Previously, we showed that 1918 NS1 populates two conformations[23]: p85β binding-competent (BC) and -incompetent (BI) conformers (Fig. 4d). Indeed, hierarchical cluster analysis showed that all 16 crystal structures could be clustered into either BI-like or BC-like conformations (Fig. 4e).

However, which of the BI and BC states represents the major conformer in the solution remained uncertain. To identify the major conformer of free NS1s, we compared NMR NOESY cross-peaks specific to either BI-like or BC-like conformers. Based on the NOESY cross-peaks pattern, we concluded that the free NS1s of the four IAV strains

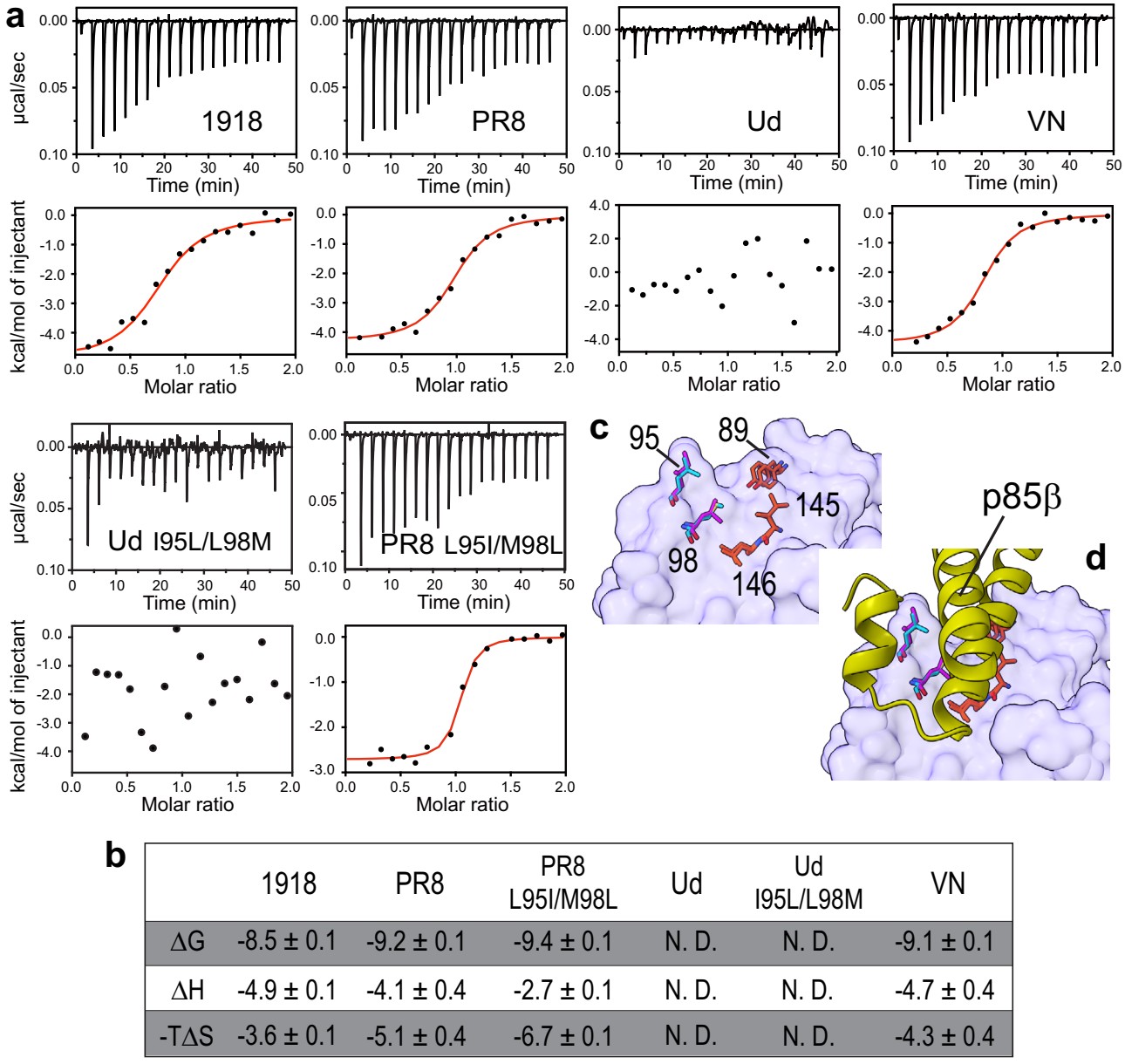

| | 1918 | PR8 | PR8 L95I/M98L | Ud | Ud I95L/L98M | VN |
|---|---|---|---|---|---|---|
| ΔG | -8.5 ± 0.1 | -9.2 ± 0.1 | -9.4 ± 0.1 | N. D. | N. D. | -9.1 ± 0.1 |
| ΔH | -4.9 ± 0.1 | -4.1 ± 0.4 | -2.7 ± 0.1 | N. D. | N. D. | -4.7 ± 0.4 |
| -TΔS | -3.6 ± 0.1 | -5.1 ± 0.4 | -6.7 ± 0.1 | N. D. | N. D. | -4.3 ± 0.4 |

**Fig. 3 | Effects of strain-specific mutations of NS1s on the thermodynamic signature for binding to p85β. a** ITC thermograms and binding isotherms for the interaction between NS1s and p85β. Red lines represent fit curves using 1:1 binding stoichiometry. **b** Estimated thermodynamic parameters for binding between NS1s and p85β. Uncertainties represent the standard deviation of three repeated measurements. N.D. represents not detected because of the absence of heat in the binding reaction. **c** Surface representation of NS1. Core interface residues are shown as sticks. Residues 95 and 98 of 1918 and Ud NS1s are shown in cyan and magenta, respectively. **d** The structure of p85β bound to NS1 is shown to illustrate the interaction with the core interface residues.

populate the BC-like conformer as the major species (Supplementary Fig. 6).

Considering the presence of the two groups of crystal structures, it is likely that the BC state of NS1 undergoes conformational exchange with the BI state. We further characterized the BI-BC conformational dynamics using the NMR $^{15}N$ CPMG relaxation dispersion[36–38]. Indeed, PR8 and VN NS1s underwent the BI-BC dynamics on a sub-ms timescale (Fig. 4f and Supplementary Fig. 7). Our previous study showed that 1918 NS1 also undergoes the BI-BC conformational dynamics on a similar timescale[23]. The analysis of slow-exchanging residues on the NMR timescale revealed that the population of the BC-like conformer was ~99% for PR8 and VN NS1s and ~90% for 1918 NS1 (Supplementary Fig. 7).

On the contrary, Ud NS1 was static and only populated the BC conformer (>99%) (Fig. 4f)[23], which is consistent with the fact that all

available Ud NS1 structures showed only a BC-like conformer. Although Ud NS1 could be dynamic in different timescales that are not detectable by NMR CPMG-RD, it was obvious that strain-specific mutations reshaped the energy landscape of free NS1s, as evidenced by altered conformational dynamics (Fig. 4f).

Given that the epistatic interactions were probed based on the mutational effects on binding rate constants, we expected that the energy landscapes of the BC state differ strain-specifically; the BC conformer is the major species and responsible for binding to p85β. Therefore, we examined the conformational variations in the BC state between NS1s.

To this end, we compared NMR chemical shifts (CS) of the backbone amide ($^{1}H$ and $^{15}N$), $^{13}C_\alpha$ and $^{13}C_\beta$ resonances in four NS1s (Fig. 5a and Supplementary Fig. 8). The root-mean-square deviation of chemical shift (RMSD-CS) among the NS1s reports the evolutionary

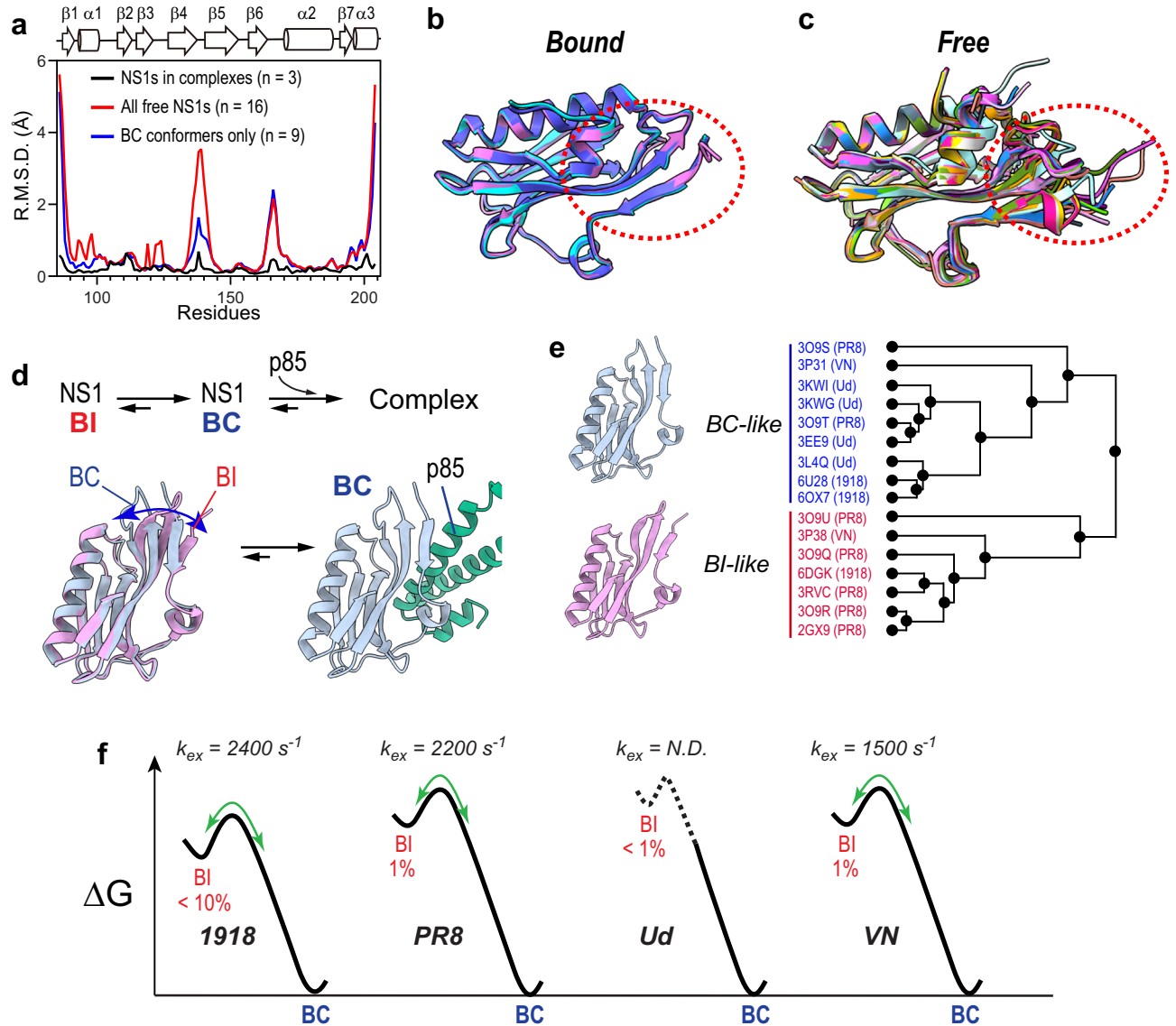

**Fig. 4 | Conformational energy landscape of NS1. a** RMSD ($n = 4$) of backbone Cartesian coordinates of NS1s. Source data are provided as a Source data file. See panel **e** for PDB IDs of NS1 structures included in the calculation. **b** Superimposed structures of 1918 (cyan), PR8 (orchid), and VN (slate blue) in complex with p85β. The structure of p85β is not shown for clarity. **c** Superimposed structures of free NS1s ($n = 13$) from 1918, PR8, Ud, and VN strains. The conformationally most heterogeneous region is marked with a red dotted circle. **d** Schematic showing the binding reaction between NS1 and p85β. BI and BC conformers are shown in orchid and pale blue, respectively. **e** The result of hierarchical cluster analysis of free NS1 structures ($n = 16$). **f** Energy landscapes illustrate the conformational exchange between BI-like and BC-like states of NS1s. Exchanging rate constants ($k_{ex}$) and population of BI conformers were determined by NMR CPMG RD (see Supplementary Fig. 7). N.D. not detected.

variation in the conformation of the BC state. In other words, conformational fluctuations of NS1 during evolution can be mapped in a site-specific manner using RMSD-CS. Residues with an RMSD-CS larger than the third quartile Q3 were identified as structurally variable residues across NS1s (Fig. 5a, b). Here, we excluded strain-specific mutations from the calculation of Q3; note that the inclusion of mutated residues altered the selection of only five residues and did not affect our analysis.

Many of the identified residues are spatially close to the mutation sites (Fig. 5a–c), indicating that the chemical shift changes represent mutational effects on the conformation of nearby residues. Moreover, the rim interface showed conformational variation across NS1s; for example, several residues in the β6-α2 loop in the rim interface showed conformational variation (Fig. 5b). This result is also consistent with the crystallographic analysis of BC conformers (Fig. 4a; blue line).

To further examine the side-chain conformational changes, we compared RMSD-CS of side-chain methyl resonances ($^1$H and $^{13}$C) of Ile, Leu, Val, Thr, Met, and Ala (ILVTMA) in four NS1s (Fig. 5d and Supplementary Figs. 9 and 10, and Supplementary Tables 2–5). Notably, all conformationally variable side-chains (i.e., RMSD-CS > Q3) clustered in the hydrophobic core (Fig. 5e). The hydrophobic core is spatially located on the opposite side of the β-sheet harboring core p85β-binding residues (Fig. 5e, f). Moreover, the hydrophobic core contains multiple strain-specific mutation sites (Fig. 5c). This result indicates that the hydrophobic core might play a key role in transmitting mutational effects to the p85β-binding interface through a long-distance network of physical contacts. Notably, none of the core interface residues showed considerable conformational variation across NS1s, while the hydrophobic core and rim interface (i.e., β6-α2 loop) were sensitive to strain-specific mutations (Fig. 5f).

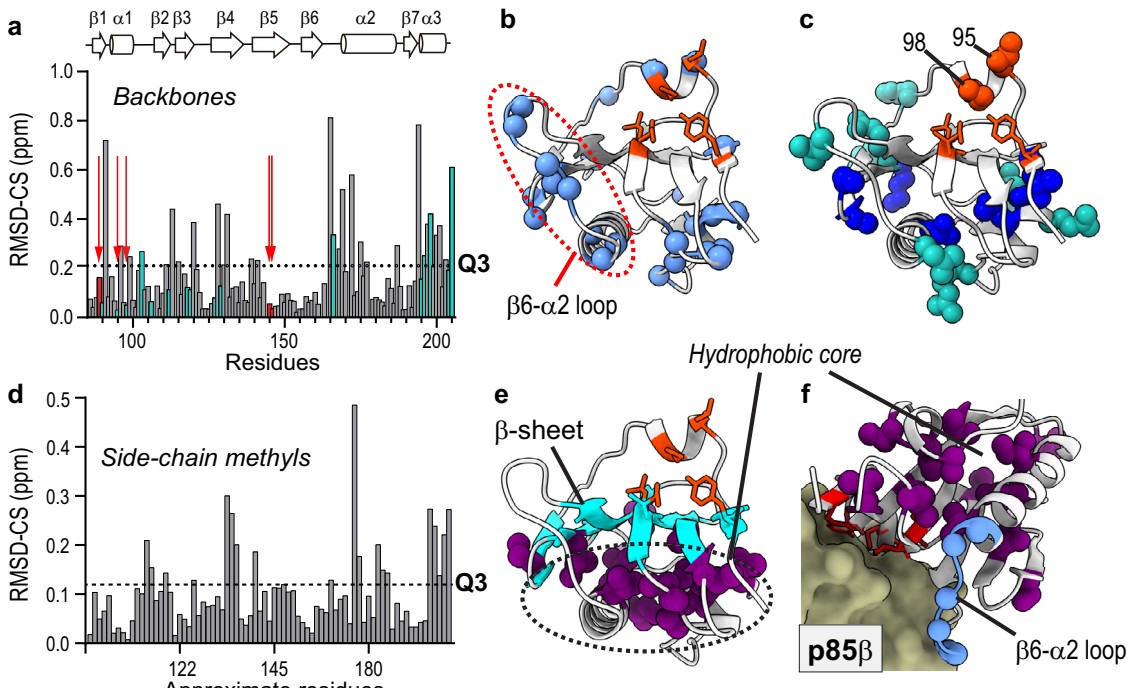

**Fig. 5 | Conformational variation across NS1s. a** RMSD-CS of backbone amides ($^{15}$NH) resonances of NS1s ($n$ = 3 or 4). Green bars correspond to strain-specific mutational sites. Dotted line represents Q3. Orange arrows show positions of core interface residues. **b** Residues whose RMSD-CS ($^{15}$NH) > Q3 are shown as light blue spheres. Core interface residues are shown as orange sticks. **c** Strain-specific mutation sites are shown as spheres. Mutations in the hydrophobic core are shown in blue spheres. **d** RMSD-CS of methyl ($^{13}$CH$_3$) resonances of NS1s. Note that *x*-axis represents approximate residue number due to space limitation. The same plot with explicit residue numbers is shown in Supplementary Fig. 10. **a, d** Source data are provided as a Source data file. **e** Residues whose RMSD-CS ($^{13}$CH$_3$) > Q3 are shown as purple spheres. The β-sheet shared by the hydrophobic core and p85-binding surface is shown in cyan. **f** Conformationally variable methyl-containing side-chains and β6-α2 loop are shown in magenta and blue spheres, respectively, in the complex with p85β (PDB ID: 6U28).

## Strain-specific mutations altered the structural dynamics of the hydrophobic core of NS1

The conformational variations in the hydrophobic core and rim interface across NS1s can result from diverse biophysical mechanisms[32,34]. Thus, we further examined the mechanistic basis of the long-range mutational effects on the rim interface.

We first tested whether the evolutionary conformational variation of NS1 is the result of changes in the overall protein stability. Long-range epistasis can be mediated by differential mutational effects on protein stability[4,11,39,40]. We measured the protein stability of NS1s using H/D-exchange coupled with NMR and the thermal shift analysis. Intriguingly, all four NS1s showed similar global protein stability (Supplementary Fig. 11), indicating that the long-range epistasis is not due to the differential stability of NS1s. These results also suggest that the stability of NS1s was already optimized during natural evolution.

Protein motion is another commonly postulated mechanism of epistatic interactions[14,15,41]; however, there is a paucity of direct experimental evidence. To test the role of protein motion in long-range epistasis, we hypothesized that mutations affected the rim interface by altering the flexibility of hydrophobic core residues of NS1. However, it should be noted that our hypothesis does not assume a contiguous pathway between a mutational site to the rim interface. Instead, we expected that the flexibility of individual residues changes differently depending on the position and nature of the mutations. For example, a comparison between strain-1 and strain-3 (Fig. 6) can show the long-range interaction between residues A and F. However, this comparison does not reveal other hidden relationships between residues E and F, for which strain-1 and strain-2 need to be compared. Moreover, the same residue may become more flexible in one NS1 but more rigid in another (Fig. 6). Taken together, studying the residue-

specific flexibility within a specific NS1 might be insufficient to identify all the residues responsible for long-range epistasis.

However, a large statistical variation in the residue-specific flexibility among NS1s indicates that the residue underwent significant changes in flexibility during NS1 evolution. Therefore, we can accept the hypothesis if the residues with a large statistical variation in flexibility are clustered in the hydrophobic core and rim interface (Fig. 6).

To characterize mutational effects on the flexibility of individual residues of NS1s, we measured the NMR order parameter ($S^2$) of the backbone amide bonds ($^1$H-$^{15}$N)[42]. Generally, NMR $S^2$ characterizes the energy landscape shaped by the conformational flexibility of individual chemical bonds. Here, $S^2$ probes the energy landscape basin corresponding to the BC-like conformer because it populates 90–99% of the entire population (Fig. 7a). For comparison, the NMR CPMG-RD probed the energy landscape shaped by conformational exchange in the slower (typically μs – ms) timescale (Fig. 7a).

Residues with a large standard deviation of $S^2$ values (i.e., SD-$S^2$ > Q3) across NS1s were identified as dynamically variable residues; that is, the conformational flexibility of these residues fluctuated widely during the evolution of NS1 (Fig. 7b and Supplementary Fig. 12). Overall, the backbone amide order parameter ($S^2_{NH}$) analysis largely recapitulated the mutational effects on the structure: the core interface residues only showed slight variation in $S^2_{NH}$ across NS1s, whereas residues with a large SD-$S^2_{NH}$ were mainly in the hydrophobic core and rim interface (Fig. 7c). Noticeably, the dynamically variable residues are mainly located at or near the mutational sites and are mainly in the well-folded secondary structures, not in the flexible loops (Fig. 7c). This result suggests that the observed variation in dynamics reflects the bona fide mutational effects and not the variation of intrinsically flexible regions by subtle changes in experimental conditions. Thus,

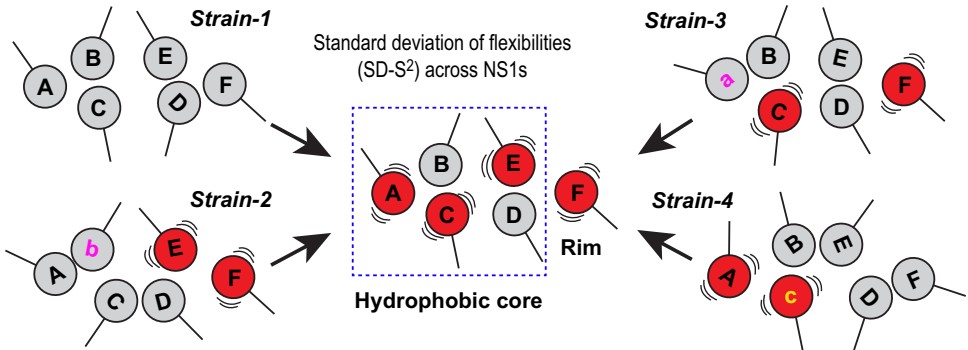

**Fig. 6 | Statistical view of long-range epistasis.** Interactions between residues A and F may not be detectable in individual NS1s, as strain-specific mutations (denoted by a, b, and c) differently perturb the flexibility of individual residues. However, residues with large changes in flexibility (red balls) during evolution can be identified by comparing the standard deviation of flexibility among NS1s. The identified residues might collectively form a network in the hydrophobic core and rim interface when they are projected onto the NS1 structure. Our hypothesis suggests that the networked residues interact with each other through direct or indirect contact during NS1 evolution.

site-specific conformational dynamics most likely varied along with the evolution of NS1.

Interestingly, three hydrophobic core residues with a dynamically variable backbone also showed conformational variations in the side-chain based on methyl chemical shift changes (Supplementary Fig. 13). Generally, however, backbone and side-chain dynamics do not necessarily correlate, especially for residues with a long side-chain[43]. So, we hypothesized that the high packing density in the hydrophobic core enabled the facile propagation of mutational effects to a longer range, and as a result, both backbone and side-chain dynamics probe mutational effects in a complementary manner.

To test the idea, we examined side-chain dynamics by measuring the methyl axis $S^2$ values based on deuterium relaxation rates in the $^{13}CH_2D$ moiety of the ILVTMA residues[44] (Supplementary Fig. 14 and 15). The dynamically variable side-chains across NS1s (SD-$S^2_{axis}$ > Q3) clustered closely in the hydrophobic core; in contrast, none of the core interface residues were dynamically variable across NS1s (Fig. 7d, e). Moreover, dynamically variable side-chains are near the residues whose backbone is dynamically variable among NS1s (Fig. 7f and Supplementary Fig. 12). For some residues, both backbone and side-chain were dynamically variable. These dynamically variable residues directly contact each other (Fig. 7f).

These observations support our hypothesis that a high packing density in the hydrophobic core enables the long-range relay of mutational effects. For example, residue 171 is the most frequently mutated position in NS1 of all human IAVs and directly contacts with residues whose side-chain or backbone showed dynamic variation during NS1 evolution (Fig. 7g). This example illustrates how strain-specific mutational effects are transmitted to the rim interface area. Although core residues make a significant energetic contribution to binding, rim residues also play a role in modulating the local environment around core residues[45–48]. Subsequently, it is likely that altered dynamics of the rim interface affect the sampling of productive conformations during the association process to p85β, resulting in strain-dependent epistasis. From the energy landscape perspective, strain-specific mutations altered the conformational energy landscape of NS1s by rearranging the packing interactions, as evidenced by the clustering of dynamically variable residues.

These results demonstrate that near-neutral mutations can silently alter the energy landscape without noticeable changes in functional phenotypes. We speculate that the cumulative effects of near-neutral mutations on the molecular energy landscape help proteins explore new functions during evolution. Therefore, understanding how mutations reshape the biophysical landscape can offer a new mechanistic basis for protein evolution and function[14,49].

## Discussion

The evolutionary inference of our results should consider the following caveats. The strain-specific mutations in NS1s employed in the present study might not be neutral with respect to in vivo viral fitness. For example, although the interaction with p85β is highly conserved, NS1s showed some differences in their interactions with other host factors[24,50]. Thus, we cannot exclude that the mutations in NS1s could affect interactions with other host factors. Here, we cautiously define the "neutral mutation" as limited to the association kinetics to p85β, not concerning other cellular or in vivo functions.

In the present study, we found long-range, strain-specific epistasis in the NS1s of human IAVs and sought to determine its mechanistic basis. Our structural study revealed that strain-specific mutations have limited impacts on the conformation of core p85β-binding residues. In contrast, the hydrophobic core and rim interface showed strain-specific conformational variations among free NS1s. Further studies revealed the hidden network of residues showing varied conformational dynamics during NS1 evolution. These networked residues structurally connect the mutational sites and the rim interface through the hydrophobic core. Therefore, our findings illustrate the high-resolution mechanism whereby a network of closely packed residues underlies long-range epistasis.

Recent studies showed that epistasis is a pervasive phenomenon in protein evolution[14,51]. A tightly packed hydrophobic core is a common structural feature of well-folded proteins. We speculate that mutations altering the packing interactions might contribute to intramolecular epistatic interactions. Thus, our study provides mechanistic insights into why epistasis prevails. This result is also related to coupled conformational dynamics in the evolution of enzyme catalysis and protein allostery[14,15,34,41,49,52,53].

It is noteworthy that the network view of dynamically variable residues only emerges by the statistical measure of fluctuations in conformational (RMSD-CS) and dynamical (SD-$S^2$) data across multiple NS1s (Fig. 6). This result indicates that the conformational or dynamic responses of individual residues in the network differ depending on the NS1 background. Indeed, this statistical picture aligns with the fluctuating pattern of long-range epistasis during NS1 evolution (Supplementary Figs. 4 and 5); strain-specific mutations alter the pattern of epistasis because one mutation remodels the packing interaction differently from another.

Our study also indicated that the molecular energy landscape of NS1s responds more sensitively to mutations during evolution than the functional landscape. For example, strain-specific mutations drastically alter the thermodynamic signature for binding to p85β in addition to affecting its structure and dynamics. Despite drastic changes in the molecular properties, the functional phenotype (i.e., association to

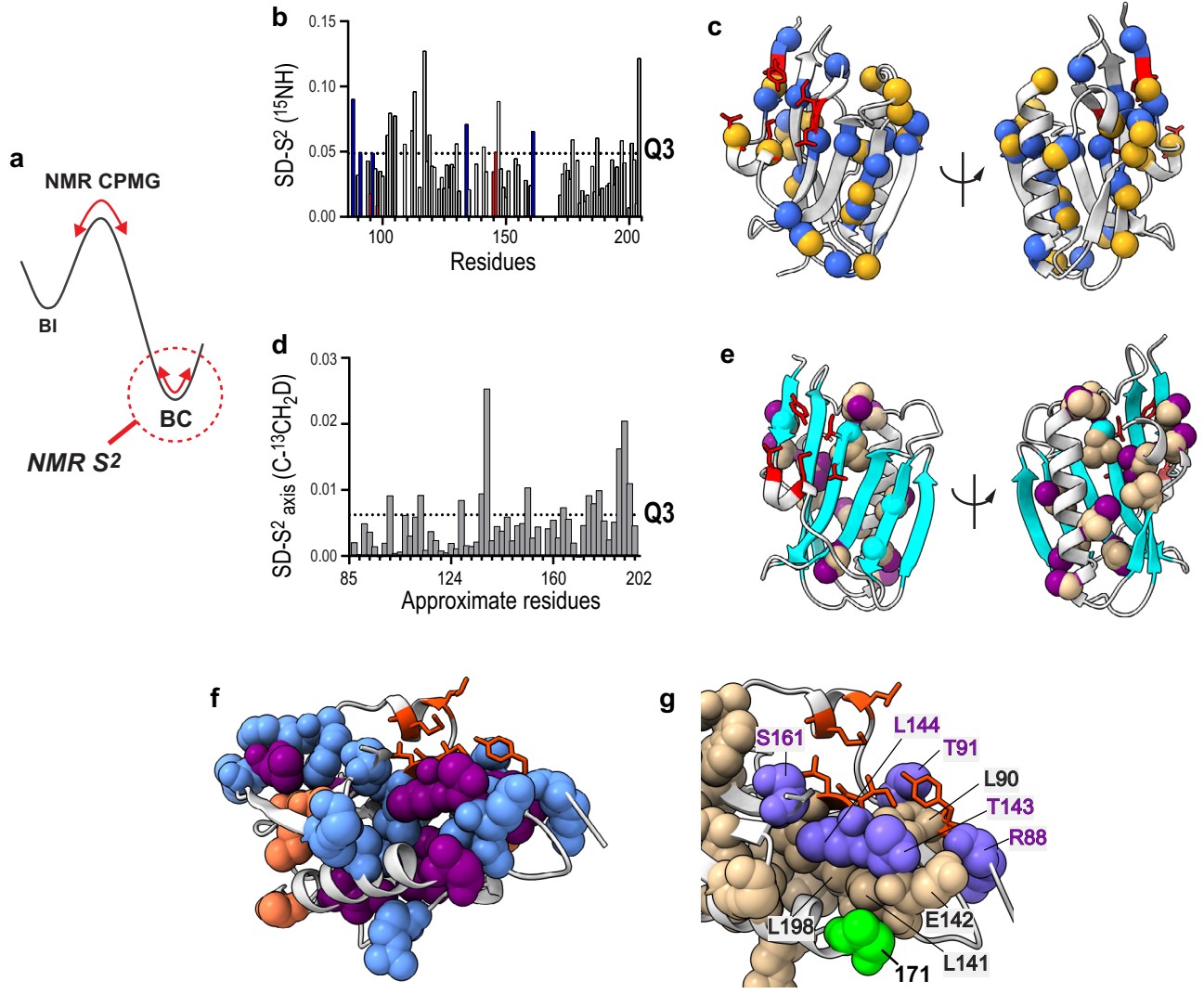

**Fig. 7 | Variation in conformational dynamics during NS1 evolution. a** Schematic showing the energy landscape mapping by NMR CPMG-RD and $S^2$ analyses. **b** Standard deviation (SD) of backbone $S^2_{NH}$ values acquired for four NS1s. Red and blue bars correspond to core and rim interface residues, respectively. Dotted line represents Q3. **c** Residues whose SD-$S^2_{NH}$ > Q3 are shown as blue spheres. Strain-specific mutational sites are shown in yellow spheres. Core interface residues are shown in red sticks. **d** SD of methyl axis order parameter ($S^2_{axis}$) acquired for four NS1s. **b, d** The exact sample size (n) for individual residues varies owing to peak overlap and variations in peak intensities. The sample sizes for individual residues are provided as a Source data file. Dotted line represents Q3. Note that x-axis corresponds to approximate residue number due to space limitation. The same

plot with explicit residue number is presented in Supplementary Figs. 14 and 15. **e** NS1 structure with residues whose SD-$S^2_{axis}$ > Q3 are shown as tan spheres. Methyl groups examined by NMR are shown in purple. β-sheet of NS1 is shown in cyan. **f** All dynamically variable residues are shown as spheres. Residues with SD-$S^2_{NH}$ > Q3 and SD-$S^2_{axis}$ > Q3 are shown in light blue and purple, respectively. Residues whose backbone and side-chain are both dynamically variable are shown in orange. **g** All dynamically variable residues are shown in spheres. Residue 171 is shown in green. Rim interface residues that are dynamically variable are shown in slate blue with residue numbers in purple. Core binding residues are shown in orange sticks. Source data are provided as a Source data file.

p85β) did not change accordingly. This result is reminiscent of thermodynamic system drift, indicating that diverse biophysical mechanisms can result in the same phenotype during protein evolution[33]. Because of the diverse hosts and high zoonotic potential of IAVs, the NS1s employed in this study are the outcome of diverse evolutionary adaptations[54]. We conjecture that strain-specific mutations allowed NS1s to explore different evolutionary pathways, resulting in various molecular properties, while the key functionality is conserved. Our results highlight the need for integrated biophysical approaches to reveal the hidden molecular foundations of protein evolution.

## Methods
### Protein sample preparation
Genes encoding all NS1 EDs and p85β proteins were prepared by gene-synthesis service from Genscript: 1918 NS1 (residues 80–205), PR8 NS1

(residues 80–206), Ud NS1 (residues 83–205), VN NS1 (residues 79–206), and p85β (residues 435–599). All NS1 proteins contain W187A mutation to prevent protein dimerization and aggregation[55]. The incorporation of W187A mutation does not interfere with the binding to p85β[23]. All proteins were expressed with the N-terminal His₆ and SUMO tags in BL21 (DE3) E. coli cells and purified by $Ni^{2+}$ NTA column chromatography. Tags were removed by SUMO protease and further purified by additional $Ni^{2+}$ NTA column and gel-filtration chromatography. All purified proteins were >95% pure; the purity of protein samples was assessed by sodium dodecyl sulfate polyacrylamide gel electrophoresis.

### NMR sample preparation
Isotopically labeled NS1 proteins for NMR studies were prepared by growing BL21 (DE3) E. coli cells in an M9 medium containing $^{15}NH_4Cl$

and $^{13}C_6$ glucose as the sole nitrogen and carbon sources. For deuterium relaxation measurements, NS1 proteins were grown in an M9 medium containing 50% $D_2O$/50% $H_2O$ with $^{15}NH_4Cl$ and $^{13}C_6$ glucose. All NMR samples were prepared in a buffer containing 20 mM sodium phosphate (pH 7), 80 mM NaCl, 1 mM TCEP, 1 mM EDTA, and 10% $D_2O$ for PR8, Ud, and VN NS1s. NMR sample for 1918 NS1 was prepared in a buffer containing 20 mM sodium phosphate (pH 7), 80 mM NaCl, 1 mM TCEP, 1 mM EDTA, 5 mM ATP, 5 mM $MgCl_2$, and 10% $D_2O$. NS1 concentration in NMR samples was 150–200 μM.

## Biolayer interferometry

The binding of surface-immobilized NS1 to p85β was measured at 25 °C using an Octet RED biolayer interferometry (Pall ForteBio). The N-terminal $His_6$ and SUMO-tagged NS1 proteins were used for immobilization. The buffer was 20 mM sodium phosphate (pH 7), 150 mM NaCl, 1% bovine serum albumin, and 0.6 M sucrose[56]. All reported values are the average and standard deviation of three repeated measurements. Association and dissociation data were fit using single exponential growth and decay functions, respectively, using GraphPad Prism (ver. 9). $K_D$ values were calculated using measured $k_{on}$ and $k_{off}$ values ($K_D = k_{off}/k_{on}$). For core interface mutants, only $k_{on}$ values were calculated by fitting a linear equation to the plot of $k_{obs}$ vs. [p85].

## NMR resonance assignment

NMR experiments were conducted on Bruker 800- and 600-MHz spectrometers equipped with a cryogenic probe at the Biomolecular NMR facility (Texas A&M University). All NMR data were collected at 25 °C. NMR data were processed using NMRPipe[57] and NMRFAM-SPARKY[58]. Sequential backbone assignments were performed using 3D triple resonance experiments, including HNCO, HN(CA)CO, HNCACB, CBCA(CO)NH, and HBHA(CO)NH. Side-chain methyl resonance assignments were performed using HCCH-TOCSY, H(CCO)NH, and C(CO)NH experiments. $^{13}C$-edited NOESY HSQC was performed with 100 ms mixing time. Chemical shifts for backbone atoms were deposited in the Biological Magnetic Resonance Bank under accession code 51403 for VN NS1 and 51404 for PR8 NS1.

## RMSD-CS

The RMSD-CS values were calculated using the following equation.

$$RMSD - CS = \sqrt{\frac{\sum(\omega_x(CS_{x,i} - <CS_x>))^2}{N}} \quad (1)$$

where $\omega_x$ represents the weighting factor accounting for the differences in the gyromagnetic ratios of different types of nuclei. CS represents a chemical shift. i represents different strains (e.g., 1918, PR8, UD, and VN), and N represents the number of NS1s included in the calculation.

## NMR $^{15}N$ Carr-Purcell-Meiboom-Gill (CPMG) relaxation dispersion (RD)

Constant relaxation time $^{15}N$ CPMG RD[36–38,59] data were recorded at 25 °C on Bruker 600- and 800-MHz NMR spectrometers with CPMG frequencies, ranging from 50 to 1000 Hz, as pseudo-three-dimensional experiments. The uncertainty of $R_{2,eff}$ was estimated by duplicated experiments. The relaxation dispersion data were fit using the Carver-Richards equation[60] for a two-state exchange model. Residues undergoing a slow-exchange process were identified based on the Akaike information criterion.

## NMR order parameters

Three relaxation parameters were measured for backbone amides of NS1s at 25 °C on Bruker 600 and 800 MHz NMR spectrometers. For $^{15}N$

$R_1$ and $R_2$ constants, five relaxation time points were taken with a recycle delay of 2 s, as previously described. For the heteronuclear nuclear Overhauser effect measurements, a recycle delay of 10 s was used in the reference experiment. The steady-state saturation of protons was performed by applying 180° pulses for 4 s[61]. $^2H$ R1 and R1ρ relaxation parameters were measured at 25 °C on a Bruker 600 MHz NMR spectrometer; pulse sequences provided by Dr. José A. Caro were used after minor modifications. For $^2H$ $R_1$, six relaxation time points were taken between 0.05 and 50 ms. For $^2H$ R1ρ, six relaxation time points were taken between 0.2 and 20 ms. Uncertainties of the relaxation parameters were estimated using duplicated measurements.

The Lipari-Szabo model-free formalism was used to calculate order parameters[42]. Overall correlation times and rotational diffusion tensors for the axially symmetric model were estimated using the program Quadric[62]. Backbone order parameters $S^2_{NH}$ were calculated using Mathematica (version 12) following the model-selection protocol and the Akaike information criterion[63]. Side-chain methyl axis order parameters $S^2_{axis}$ were calculated using the following equation[64].

$$J(\omega) = (2/5)[S^2\tau_m/(1+(\omega\tau_m)^2) + (1-S^2)\tau_i/(1+(\omega\tau_i)^2)] \quad (2)$$

where $S^2$ is an order parameter for the methyl group, $\tau_m$ is the overall correlation time, and $\tau_i^{-1} = \tau_m^{-1} + \tau_{e,i}^{-1}$ with $\tau_{e,i}$ is the effective correlation time of the internal motions for the $^{13}C$-$^2H$ bond vector. $S^2_{axis}$ was calculated by $S^2 = 0.111 S^2_{axis}$, assuming tetrahedral geometry for the methyl group[64].

$S^2_{axis}$ values of Leu $^{13}C^\delta$, whose chemical shifts are larger than 24.5 ppm, were eliminated to prevent artifacts arising from a strong $^{13}C$-$^{13}C$ coupling on $S^2_{axis}$[65].

## NMR H/D exchange

Buffer containing 150 μM $^{15}N$ labeled NS1 was exchanged against 20 mM sodium phosphate (pD 7), 80 mM NaCl, 1 mM TCEP, and 1 mM EDTA in 99.9% deuterium oxide using a spin desalting column (Micro Bio-Spin 6, BioRad). $^{15}N$-HSQC spectra were measured every 40 min for 106 h at 25 °C on Bruker 600 MHz NMR spectrometer. H/D exchange rates of backbone amide resonances of NS1 were calculated from a single exponential fit. Protection factors were calculated from the H/D exchange rate divided by the predicted intrinsic exchange rate from the program SPHERE[66].

## X-ray crystallography

The VN NS1 W187A:p85β complex was crystallized at 277 K by sitting-drop vapor diffusion in 20 mM Tris (pH 7.0) and 80 mM NaCl. The crystal was flash-frozen in liquid nitrogen in the reservoir solution containing 25% (v/v) glycerol. X-ray diffraction datasets were collected at 120 K using an R-AXIS IV++ image plate detector mounted on a Rigaku MicroMax 007HF X-ray generator. The data were processed using iMosflm in the CCP4 package[67]. The structure was remodeled and refined with Coot[68] and the Phenix package (Supplementary Table 1). The crystal structure of VN NS1 W187A in a complex with p85β is deposited in the PDB (ID code: 7RCH). Protein structures were visualized using Pymol and ChimeraX 1.4[69].

## ITC measurements

All ITC samples were prepared in 20 mM sodium phosphate (pH 7), 100 mM NaCl, and 2 mM TCEP. Data were recorded at 25 °C using a Microcal-PEAQ-ITC calorimeter (Malvern Panalytical). 100 μM p85β was in the syringe, and 10 μM NS1 was in the cell. 19 consecutive 2 μl aliquots of p85β were titrated into the cell. Data were fit using a 1:1 binding model to obtain $K_a$, $\Delta H$, and $\Delta S$ using software provided with

the instrument. All reported parameters are the average and standard deviation of two repeated measurements.

## Thermal shift analysis

Protein melting temperatures ($T_m$) were measured using thermal shift analysis. 0.1 μM NS1 was mixed with Sypro Orange (S5692, Sigma-Aldrich) in 20 mM sodium phosphate (pH 7), 100 mM NaCl, and 2 mM TCEP. Data were recorded from 20 °C to 80 °C using a Quantamaster 400 fluorometer (Photon Technology International). $T_m$ was calculated by fitting data to Boltzman sigmoid equation (GraphPad Prism).

## NS1 sequence comparison

The NS1 sequences were aligned using Clustal Omega[70] and the sequence logo was generated using WebLogo[71].

## Reporting summary

Further information on research design is available in the Nature Research Reporting Summary linked to this article.

## Data availability

All data generated in this study are available within the Article and Supplementary information. The coordinate of VN NS1 in complex with p85β are available in the Protein Data Bank (PDB) under accession code 7RCH. NMR chemical shifts for backbone atoms are available in the Biological Magnetic Resonance Bank under accession code 51403 for VN NS1 and 51404 for PR8 NS1. Source data are provided with this paper.

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

## Acknowledgements

We thank Prof. Wonmuk Hwang, Prof. Tatyana Igumenova, and James Gonzales for carefully reading the manuscript and providing constructive comments. Support from NIH grant R01GM127723 (J.H.C.), the Welch Foundation A-2028-20200401 (J.H.C.), and USDA National Institute of Food and Agriculture grant Hatch project 1020344 (J.H.C.) is acknowledged.

## Author contributions

J.H.C. conceived the project. I.K. conducted NMR resonance assignments and ITC measurements. A.D. and B.Zu. acquired BLI data. B.Zh., I.K., P.L., and J.H.C. conducted protein crystallization and structure determination. N.S. and A.B. conducted co-IP.

## Competing interests

The authors declare no competing interests.
