## [Peer Review File · Nature Communications]

Reviewer comments, first round review:

Reviewer #1 (Remarks to the Author):

In this manuscript Kim et al. investigate the mechanistic basis for epistasis in four constructs of the NS1 protein of the influenza A virus (referred to as 1918, PR8, VN, and Ud). The four constructs have strain-specific mutations localized within the core. On the contrary, the residues forming the binding interface with p85 β are conserved among the four strains. All four NS1 constructs share the same binding kinetics (k_{on} and k_{off}) to p85 β . Using alanine-scanning, the authors show that single-point NS1 mutations at the interface perturb the k_{on} differently for the different strains, indicating the existence of epistatic interactions between the background and the interface residues. To further address the role of the background residues (i.e., the NS1 residues not directly involved in the interaction with p85 β) in determining the activity of the different NS1s, the authors acquire ITC data showing that while binding of Ud to p85 β is entropically driven, binding of 1918, PR8 and VN to p85 β is enthalpically driven. These data further support the idea that background residues affect the biophysical mechanism that the different NS1 proteins utilize to achieve the same function (i.e. binding to p85 β with identical binding kinetics).

To address the basis for the above observation, the authors compare the X-ray structures of the free and complexed NS1 proteins (the structure of the Ud-p85 β complex is not available) and determine that, since the structures of the bound NS1s are identical, epistasis must originate from the difference in structure and dynamics of the free state. Analyzing the NMR chemical shifts, they show the existence of conformational variations among the different NS1 strains that clustered in the core region (where most of the strain-specific mutations localize) and at the rim interface. Analyzing the NMR-derived order parameters, they show that variation in dynamics across the strains are also localized in the same regions. Based on this evidence, the authors suggest that "altered dynamics of the rim interface affect the sampling of productive conformations during the association process to p85 β , resulting in strain-dependent epistasis."

Overall, I have enjoyed reading this manuscript. The data seem to be acquired and analyzed rigorously. However, it is my opinion that they are not sufficient to prove unambiguously the authors' hypothesis. My main objection is that, in the absence of additional data on the conformational dynamics of the NS1s bound to p85 β it cannot be excluded that epistasis originates from modulation of the dynamics of the bound state. In addition, it is not clear from the present form of the manuscript how, in the absence of a network of residues that transmits the signal, the authors physically connect the changes in protein dynamics within the core with the once observed at the rim interface. Therefore, I think the manuscript is not ready for publication in the present form. Here are some suggestions to address my concerns.

-The authors should compare the dynamics of the NS1s in their bound state by NMR (the ideal methods, but probably challenging due to boarder-line size of the NS1-p85 β complex) or MD simulations.

-The authors should propose a physical mechanism that underlies communication between the core and the rim

-They should better define the criteria used to infer the absence of the BI conformation from the analysis of the NOE experiments. In other words, they used NOEs that are observed only for the BC conformation to detect its presence in solution. Which NOEs that are only observed in the BI conformation were used to prove that BI is sampled at a lesser extent in solution?

-Please perform the RMSD-CS analysis also on the secondary C α and C β chemical shifts, which are more accurate reporters of structural changes than the amide chemical shifts.

-In Supplementary Figure 11, please provide the LogPF versus residue

-Please report the melting temperatures for the four NS1 constructs to support the conclusion that they all have the same stability

- Please plot the secondary structure on top the S2 values in Supplementary Figure 12.
- Please define "sign epistasis" in the same way you have defined positive and negative epistasis
- The authors say they observe positive, negative and sign epistasis. However, in Supplementary Figure 5 I don't see any plot labeled as positive epistasis.
- Please report the equation used for calculating the RMSD-CS in methods

Reviewer #2 (Remarks to the Author):

Kim et al. initiated the study by measuring the binding kinetics between p85 β and NS1s from 4 different influenza A strains (1918, PR8, Ud, VN), and performing mutagenesis experiments. They showed that introducing the same mutation into different NS1s could result in very different effects on p85 β binding kinetics, illustrating the existence of epistasis. The authors also showed that Ud NS1 was very different from the other 3 NS1s in that the binding to p85 β was not accompanied by measurable heat and was driven exclusively by entropy. Subsequently, extensive NMR analysis indicated that the structural dynamics of the NS1 hydrophobic core have evolved over time and contributed to epistasis.

Overall, this study provides important biophysical insights into protein evolution, epistasis, and influenza biology. However, there are some concerns about the interpretation of the data (see below).

Major comments:

1. In the methods section: "Association and dissociation data [from biolayer interferometry] were fit using single exponential growth and decay functions". However, the goodness-of-fit (R-squared) is not reported. In Supplementary Figures 2 and 3, the R-squared needs to be shown for each panel. Showing the R-squared is especially important, since several conclusions in this study highly rely on the model fitting of biolayer interferometry data.
2. The explanation for the statistical view of long-range epistasis can be further clarified. For example, it seems like we can conclude that there is a long-range interaction between A and F by comparing Strain-1 vs Strain-3, which contains a mutation at residue A. So is it really necessary to analyze all strains?
3. Related to the previous comment, can the authors clarify what is meant by "directly involved" and "indirectly involved" in this sentence: "Our hypothesis suggests that the networked residues are directly or indirectly involved in the long-range interaction..." (legend of Figure 6)?
4. The authors claimed that "Notably, none of the core interface residues showed considerable conformational variation across NS1s". However, according to Figures 5a and 5b, it seems like residue 98, which is a core interface residue, has an RMSD-CS (15NH) > Q3?
5. The authors state that: "many hydrophobic core residues with a dynamically variable backbone also showed conformational variations in the side-chain based on methyl chemical shift changes (Supplementary Fig. 13)." However, accordingly to Supplementary Figure 13, it seems like there are only 3 residues with both dynamically variable backbone and conformationally variable side chain. So the word "many" seems inappropriate?
6. Does the NMR data provide insight into why the binding of Ud NS1 to p85 β is exclusively driven by entropy?
7. Similarly, does the NMR data provide insight into why mutations at the core interface residues have the least deleterious effect for Ud NS1 (Figure 2c-d)?

Minor comments:

1. In Supplementary Fig. 1a, it will be helpful to indicate what is the % sequence identity of NS1 between each strain and the 1918 strain.
2. In Supplementary Fig. 1a, it will also be helpful to indicate the locations of the core interface residues 89, 95, 98, 145, and 146.
3. In the second paragraph of the results section, is “dissociation rate constant” equivalent to “KD” in Figure 1c?
4. The error bars in Figures 2c-d and Supplementary Figure 4 need to be defined in the figure legends.
5. In Figure 4e, it will be useful to label the strain name next to the PDB ID to support the statement: “all available Ud NS1 structures showed only a BC-like conformer.”
6. Does the conformationally most heterogeneous region (marked with a red dotted circle in Figure 4c) overlap with the p85 β binding site?

REVIEWER COMMENTS

Reviewer #1 (Remarks to the Author):

In this manuscript Kim et al. investigate the mechanistic basis for epistasis in four constructs of the NS1 protein of the influenza A virus (referred to as 1918, PR8, VN, and Ud). The four constructs have strain-specific mutations localized within the core. On the contrary, the residues forming the binding interface with p85 β are conserved among the four strains. All four NS1 constructs share the same binding kinetics (k_{on} and k_{off}) to p85 β . Using alanine-scanning, the authors show that single-point NS1 mutations at the interface perturb the k_{on} differently for the different strains, indicating the existence of epistatic interactions between the background and the interface residues. To further address the role of the background residues (i.e., the NS1 residues not directly involved in the interaction with p85 β) in determining the activity of the different NS1s, the authors acquire ITC data showing that while binding of Ud to p85 β is entropically driven, binding of 1918, PR8 and VN to p85 β is enthalpically driven. These data further support the idea that background residues affect the biophysical mechanism that the different NS1 proteins utilize to achieve the same function (i.e. binding to p85 β with identical binding kinetics).

To address the basis for the above observation, the authors compare the X-ray structures of the free and complexed NS1 proteins (the structure of the Ud-p85 β complex is not available) and determine that, since the structures of the bound NS1s are identical, epistasis must originate from the difference in structure and dynamics of the free state. Analyzing the NMR chemical shifts, they show the existence of conformational variations among the different NS1 strains that clustered in the core region (where most of the strain-specific mutations localize) and at the rim interface. Analyzing the NMR-derived order parameters, they show that variation in dynamics across the strains are also localized in the same regions. Based on this evidence, the authors suggest that "altered dynamics of the rim interface affect the sampling of productive conformations during the association process to p85 β , resulting in strain-dependent epistasis." Overall, I have enjoyed reading this manuscript. The data seem to be acquired and analyzed rigorously. However, it is my opinion that they are not sufficient to proof unambiguously the authors' hypothesis. My main objection is that, in the absence of additional data on the conformational dynamics of the NS1s bound to p85 β it cannot be excluded that epistasis originates from modulation of the dynamics of the bound state. In addition, it is not clear from the present form of the manuscript how, in the absence of a network of residues that transmits the signal, the authors physically connect the changes in protein dynamics within the core with the once observed at the rim interface. Therefore, I think the manuscript is not ready for publication in the present form. Here are some suggestions to address my concerns.

We thank the reviewer for reading our manuscript carefully. The reviewer's comments and questions helped us revise the manuscript. We provide our response to the reviewer's concerns below.

The reviewer commented on the role of the bound state in the observed epistasis of NS1. First, we would like to clarify that we identified epistasis in NS1 based on binding rate constants (k_{on} values), not based on binding affinities (K_D values). Thus, observed epistasis (Fig. 2 and Supplementary Fig. 4) is not related to the bound state. In other words, our analysis focused on mutational effects on the activation barrier for binding (Figure 2b shows the concept behind our approach). Moreover, the timescale of conformational dynamics probed by NMR is substantially slower than the lifetime of the transition state; thus, the effect of conformational dynamics on the transition state must be minimal, if indeed there are any. Taken together, we reached the conclusion that strain-specific epistatic effects are primarily due to the heterogeneity among free NS1s. We hope that we have addressed the reviewer's concern.

-The authors should compare the dynamics of the NS1s in their bound state by NMR (the ideal methods, but probably challenging due to boarder-line size of the NS1-p85 β complex) or MD simulations.

This suggestion is related to the reviewer's comment above. Although the NS1 dynamics in the bound state are not related to the observed epistasis (see our response above), we agree with the reviewer that a comparison of the NS1 dynamics between the free and bound states can provide additional insight into the strain-specific thermodynamic energy landscape probed by ITC. We presented the ITC data because it unambiguously shows that strain-specific mutations altered the thermodynamic energy landscape of the NS1-p85 β interaction. Although this conclusion is clear, studies on both free and bound states are required to determine the mechanism underlying this difference.

Unfortunately, however, we were unable to study the bound state owing to the extremely low solubility of the NS1-p85 complex. For example, the complex with 10 μ M NS1 saturated with 30 μ M p85 β was only soluble for < 2 h after mixing (see figure on the right; 1D methyl TROSY spectra of the NS1-p85 β mixture).

To overcome the low solubility, we have applied a number of approaches, such as adding multiple types of solubility tags or small protein domains in either the N- or C-terminus of NS1 or p85 β , and by incorporating solubility-enhancing solutes, such as the glutamate-arginine mixture, ATP, trehalose, and sucrose. None of the trials made the complex sufficiently soluble for NMR experiments. Ironically, the low solubility enabled the formation of crystals of the complexes, and all the crystals of the complexes were formed from precipitated proteins. As the reviewer pointed out, the NMR study of the complex is challenging because of its large size, even without the solubility issue. Considering the poor solubility of the complex even at \sim 10 μ M, the NMR study of the complex was not feasible. We also agree with the reviewer that MD simulations may be an alternative approach for the analysis of complexes; however, our lab currently does not have the capacity to perform MD simulations. In the future, we will pursue the approach through collaboration.

Considering the challenging conditions for the complex, we decided to pursue an alternative approach using crystallographic B-factors that can provide information on protein flexibility under crystalline conditions. However, B-factors are affected not only by protein flexibility but also by crystal defects, resolution, and refinement strategies (Acta Crystallogr D Struct Biol, 2022, vol.78, 69-74). Although effects of factors other than protein flexibility can be reduced by comparing normalized B-factors (B_{Norm}), it should be noted that comparison of B_{Norm} is only valid for structures with a similar resolution. Because this condition was not met by our collection of crystal structures, we limited our analysis to gaining a simple insight into the difference in the overall dynamics of NS1 between the free and bound states.

The figure on the right side shows the standard deviations of the B_{Norm} values for individual residues in the free (i.e., BC conformers) and p85-bound NS1s. These results indicate that individual residues of NS1 in the bound state are accompanied by smaller standard

deviations for B_{Norm} values than those in the free state. Hence, the flexibility variation across NS1s is substantially reduced in the bound state compared to the free state. In other words, the strain-specific variation of NS1 dynamics in the bound state is less influential, if indeed there are any, on the ITC data compared to the effects of the free state. Although this result is consistent with our interpretation of the data in the manuscript, we are aware of the limitations associated with the B-factor-based approach and, at this point, are unable to speculate further on the NS1 dynamics in the bound state.

-The authors should propose a physical mechanism that underlies communication between the core and the rim

The rim and core interface residues cooperatively contribute to the binding kinetics and thermodynamics. Typically, core residues (often called hot spot residues) are hydrophobic and aromatic residues, as also shown at the NS1-p85 β interface (Proteins, 2002, vol. 47, 334-343). Although core residues are major contributors to binding energetics, the surrounding residues (i.e., rim) need to sequester the hydrophobic interface. Thus, the local environment of the binding interface can be modulated by the structure and dynamics of surrounding residues.

For example, NS1-Y89 is a highly conserved core interface residue. It forms a buried hydrogen bond to p85-D575; this hydrogen bond is the most energetically significant interaction between NS1 and p85 β . This type of buried ionic/polar interaction is frequently found in the protein-protein interface (JMB, 2005, Vol.345, 1281-1294). Significantly, the strength of the buried hydrogen bond depends on the local hydrophobic environment, which is modulated by the structure and dynamics of the surrounding residues (PNAS, 2020, vol. 117, 6550-6558; Protein Sci, 2014, vol 23, 652-661). Moreover, the rim or periphery of a binding interface can affect the protein-protein interactions by modulating the electrostatic environment (PNAS, 2004, vol. 101, 9223-9228; NSB, 2000, vol. 7, 537-541).

Our study showed that the conformational dynamics of the core interface residues did not vary across NS1 evolution, whereas the rim residues showed varying dynamics. The differential dynamics of the rim residues variously affect the local hydrophobic environment around the core residues. Consequently, rim residues can affect the energetics of the core residues. As the reviewer suggested, we added the following sentence with the references referred to above about rim-core interactions in the revised manuscript (page 16):

"Although core residues make a significant energetic contribution to binding, rim residues also play a role in modulating the local environment around core residues^{45,46,47,48}."

-They should better define the criteria used to infer the absence of the BI conformation from the analysis of the NOE experiments. In other words, they used NOEs that are observed only for the BC conformation to detect its presence in solution. Which NOEs that are only observed in the BI conformation were used to prove that BI is sampled at a lesser extent in solution?

To identify the major conformation of NS1 in the free state, we compared pairwise distances that were less than 5 Å in either the BI or BC conformation, but not in both conformers. Thus, the presence of NOE cross-peaks corresponding to one conformer excludes the other unless both conformers are similarly populated. Overall, we included 5-7 NOESY cross-peaks for identifying the conformations of individual NS1s (see table below). The original Supplementary Figure 6 shows the representative NOESY cross-peaks included in the conformational identification and

their positions in the BI and BC structures. To further clarify our procedure, we added a list of pairwise distances included in our conformational evaluation of individual NS1s (Supplementary Figure 6).

1918				PR8			
Atom1	Atom2	Distance		Atom1	Atom2	Distance	
		BI (Å)	BC (Å)			BI (Å)	BC (Å)
L90 HD	F134 HD	7.0	4.3	L90 HD	F201 HE	8.9	3.1
L90 HD	F201 HD	9.5	4.5	V136 HG	F201HD	7.3	3.2
V136 HG	F201 HB	7.8	3.9	V136 HG	F201 HE	7.8	3.3
L141 HD	V174 HB	8.6	2.6	L141 HD	A171 HA	8.3	2.5
L141 HD	V174 HG	7.0	2.7	L141 HD	A171 HB	7.5	3.3
L141 HD	A202 HB	7.7	2.1	L141 HD	A202 HB	8.5	3.6

Ud				VN			
Atom1	Atom2	Distance		Atom1	Atom2	Distance	
		BI (Å)	BC (Å)			BI (Å)	BC (Å)
I90 HD1	F201 HE	NA	3.8	L90 HD	F201 HD	7.7	2.8
I90 HG2	F201 HE	NA	3.4	L90 HD	F201 HE	12.8	3.1
V136 HG	F201 HB	NA	2.6	V136 HG	F201 HD	8.6	3.2
V136 HG	F201 HD	NA	2.8	L141 HD	G171 HA	9.3	2.9
V136 HG	F201 HE	NA	3.4	L141 HD	V174 HG	7.2	2.4
L141 HD	I171 HA	NA	2.8				
L141 HD	A202 HB	NA	2.7				

NA: Not available.

-Please perform the RMSD-CS analysis also on the secondary C α and C β chemical shifts, which are more accurate reporters of structural changes than the amide chemical shifts.

We thank the reviewer for this suggestion. We added the new data as Supplementary Figure 8b. Briefly, the analysis of RMSD-CS using combined C α and C β chemical shifts resulted in a pattern similar to that obtained using backbone amide resonances (see figure on the right). This data further supports our approach to identifying conformationally variable regions across NS1 evolution.

Additionally, we replaced the original Figures 5a and 5b with modified Figures 5a and 5b because we found during the revision that the original version was prepared using data including chemical shifts of mutated residues. NMR chemical shifts change according to the identity of the amino acids, in addition to conformational changes. Thus, the chemical shift values of the mutated residues had to be excluded from the calculation, but they were included in the original Figures 5a-b. Although this modification did not affect our conclusion, we sincerely apologize for this mistake.

-In Supplementary Figure 11, please provide the LogPF versus residue

We added the LogPF as a function of residue in Supplementary Figure 11.

-Please report the melting temperatures for the four NS1 constructs to support the conclusion that they all have the same stability

We added the new data on the melting temperature (T_m) of NS1s to Supplementary Figure 11. Briefly, this additional data is consistent with the result of the NMR-derived stability analysis (see attached figure below). Thus, the data further support our conclusion that the strain-specific epistasis is not due to the stability difference.

-Please plot the secondary structure on top the S2 values in Supplementary Figure 12. We added the secondary structure diagram.

-Please define "sign epistasis" in the same way you have defined positive and negative epistasis

Sign epistasis means that the effect of a double mutation is beneficial relative to one type of single mutation and deleterious relative to another single mutation. For a graphical explanation of sign epistasis, we show modified Figure 2e on the right. Here, ab^* represents the $\Delta\Delta G_{on}$ of the double mutant, while ab (without *) is the calculated $\Delta\Delta G_{on}$ in the absence of epistasis (i.e., pure additive). The effect of the double mutation ab^* is beneficial (i.e., less deleterious) with respect to a single mutation Ab (shown in green), whereas it is deleterious with respect to another single mutation, aB (shown in blue). Following the reviewer's suggestion, we added the following sentences to the revised manuscript (page 6):

"Sign epistasis means that the effect of a double mutation is beneficial with respect to one type of single mutation background and deleterious with respect to another single mutation background."

-The authors say they observe positive, negative and sign epistasis. However, in Supplementary Figure 5 I don't see any plot labeled as positive epistasis.

We apologize for the typo. We meant "additive," not "positive." We corrected it in the revised manuscript. We thank the reviewer for carefully reading the manuscript and

-Please report the equation used for calculating the RMSD-CS in methods

We added the equation for calculating the RMSD-CS in the Methods section. Briefly, it was calculated using the following equation:

$$\text{RMSD-CS} = \sqrt{\sum (\omega_x (CS_{x,i} - \langle CS_x \rangle))^2 / N}$$

where ω_x represents the weighting factor accounting for the differences in the gyromagnetic ratios of different types of nuclei. CS represents chemical shifts. i represents different strains (e.g., 1918, PR8, UD, and VN in our research), and N represents the number of NS1s included in the calculation.

Reviewer #2 (Remarks to the Author):

Kim et al. initiated the study by measuring the binding kinetics between p85 β and NS1s from 4 different influenza A strains (1918, PR8, Ud, VN), and performing mutagenesis experiments. They showed that introducing the same mutation into different NS1s could result in very different effects on p85 β binding kinetics, illustrating the existence of epistasis. The authors also showed that Ud NS1 was very different from the other 3 NS1s in that the binding to p85 β was not accompanied by measurable heat and was driven exclusively by entropy. Subsequently, extensive NMR analysis indicated that the structural dynamics of the NS1 hydrophobic core have evolved over time and contributed to epistasis.

Overall, this study provides important biophysical insights into protein evolution, epistasis, and influenza biology. However, there are some concerns about the interpretation of the data (see below).

We thank the reviewer for their positive comments and insightful questions. We addressed the reviewer's concerns and suggestions below.

Major comments:

1. In the methods section: "Association and dissociation data [from biolayer interferometry] were fit using single exponential growth and decay functions". However, the goodness-of-fit (R-squared) is not reported. In Supplementary Figures 2 and 3, the R-squared needs to be shown for each panel. Showing the R-squared is especially important, since several conclusions in this study highly rely on the model fitting of biolayer interferometry data.

We thank the reviewer for their careful review of our manuscript. We added the root mean sum-of-squared errors ($S_{y,x}$) to show the goodness-of-fit of the nonlinear model fit (i.e., exponential fitting to data). Please note that the R^2 value is only valid for the linear model fit and not for the nonlinear model fit (BMC Pharmacology, 2010, vol. 10, 6). $S_{y,x}$ is defined as follows:

$$S_{y,x} = \sqrt{(\sum residual^2)/(n - K)}$$

where residual is the vertical distance of the data from the fit curve, and n and K are the number of data and fitting parameters, respectively. The unit of $S_{y,x}$ is the same as that for the Y axis. In addition, we added R^2 values for the linear plot of k_{obs} vs. [p85], as the reviewer suggested.

2. The explanation for the statistical view of long-range epistasis can be further clarified. For example, it seems like we can conclude that there is a long-range interaction between A and F by comparing Strain-1 vs Strain-3, which contains a mutation at residue A. So is it really necessary to analyze all strains?

We thank the reviewer for this suggestion. Following the example used by the reviewer, the comparison of strain-1 and strain-3 will show the long-range interaction between residues A and F (see figure below). However, this comparison does not show other hidden relationships between residues E and F, for which strain-1 and strain-2 need to be compared. Similarly, the relationship between residues C and E would not be evident unless the results for all strains are included. Thus, to reveal the hidden networks, it may be necessary to include multiple NS1s.

As the reviewer suggested, we added the following description to the revised manuscript (page 13):

"For example, a comparison between strain-1 and strain-3 (Fig. 6) can show the long-range interaction between residues A and F. However, this comparison does not reveal other hidden relationships between residues E and F, for which strain-1 and strain-2 need to be compared."

3. Related to the previous comment, can the authors clarify what is meant by "directly involved" and "indirectly involved" in this sentence: "Our hypothesis suggests that the networked residues are directly or indirectly involved in the long-range interaction..." (legend of Figure 6)?

If a mutation affects the dynamics of neighboring residues through direct physical contact, then they are directly involved. For example, Figure 6 (see figure above) shows that residue A can "directly interact" with residue C, enabling the direct transmission of mutational effects between the two. In comparison, residue A does not directly interact with residues D and E.

Nevertheless, mutation of residue A (strain 3) can "indirectly" alter the conformations and/or dynamics of residues D and E, resulting in altered dynamics for residue F. Therefore, residues D and E are "indirectly involved" in the network underlying the long-range interaction between residues A and F. As the reviewer suggested, we added the following revised sentence to the legend for Figure 6 (page 13):

"Our hypothesis suggests that the networked residues interact with each other through direct or indirect contact during NS1 evolution."

4. The authors claimed that "Notably, none of the core interface residues showed considerable conformational variation across NS1s". However, according to Figures 5a and 5b, it seems like residue 98, which is a core interface residue, has an RMSD-CS (15NH) > Q3?

We thank the reviewer for carefully reviewing the data. During the calculation of RMSD-CS (¹⁵NH), we excluded the chemical shifts associated with mutated residues. NMR chemical shifts change according to the identity of the amino acids, in addition to conformational changes. Thus, the chemical shifts of the mutated residues were excluded from the calculation. However, we found that the original Figures 5a-5b were prepared using the RMSD-CS values that included ones for mutated residues. Accordingly, we revised Figures 5a-5b using the RMSD-CS without the chemical shifts of mutated residues. Based on the correct RMSD-CS values, we

confirmed that none of the core interface residues had RMSD-CS (^{15}NH) > Q3. We sincerely apologize for this mistake and thank the reviewer again for carefully reviewing our manuscript.

5. The authors state that: "many hydrophobic core residues with a dynamically variable backbone also showed conformational variations in the side-chain based on methyl chemical shift changes (Supplementary Fig. 13)." However, accordingly to Supplementary Figure 13, it seems like there are only 3 residues with both dynamically variable backbone and conformationally variable side chain. So the word "many" seems inappropriate?

We agree with the reviewer. We corrected the sentence as follows (page 14):

"Interestingly, **three** hydrophobic core residues with a dynamically variable backbone also showed conformational variations in the side-chain, based on methyl chemical shift changes (Supplementary Fig. 13)."

6. Does the NMR data provide insight into why the binding of Ud NS1 to p85 β is exclusively driven by entropy?

We thank the reviewer for this question, and we share the reviewer's interest in the origin of the entropy-driven binding of Ud NS1 to p85b. It is now well accepted that conformational entropy change in proteins can substantially contribute to the total binding entropy. Briefly, the total binding entropy measured by ITC is the sum of the changes in the conformational entropy of proteins and solvents and the rotational-diffusion entropy (PNAS, 2017, vol. 114, 6563-6568). The conformational entropy ($-\Delta S_{\text{conf}}$) can be estimated by measuring the change in the NMR order parameters of NS1s between the free and bound states, $-\Delta S^2$ (bound – free).

Thus, we sought to address this question by measuring the order parameters of NS1s in the free and bound states. Unfortunately, however, we were unable to conduct any NMR experiments on the bound state because of its extremely poor solubility (please see our response to reviewer 1). Thus, we can only hypothesize that Ud NS1 is accompanied by a lower conformational entropy penalty upon binding to p85 β because Ud NS1 has a more rigid p85 β -binding interface than other NS1s. To test this hypothesis, we compared the NMR order parameters (S^2_{axis}) of methyl-containing residues at the p85 β -binding interface, including both the core and rim residues. Indeed, we found that Ud NS1 has a higher average S^2_{axis} value (i.e., more rigid) at the p85 β -binding interface compared to other NS1s (see table below).

NS1s	S^2_{axis}	Included methyl groups
1918	0.44 ± 0.26	91- γ , 93- ϵ , 95- $\delta 2$, 98- ϵ , 143- γ , 145- δ , 146- $\delta 1$
PR8	0.49 ± 0.26	91- γ , 93- ϵ , 98- ϵ , 143- γ , 144- $\delta 1$, 145- δ , 146- $\delta 2$
Ud	0.61 ± 0.18	91- γ , 93- ϵ , 95- $\gamma 2$, 143- γ , 144- $\delta 1$, 145- γ , 146- $\delta 1$
VN	0.44 ± 0.30	91- γ , 93- ϵ , 95- $\delta 1$, 95- $\delta 2$, 98- ϵ , 143- γ , 144- $\delta 1$, 145- δ , 145- γ , 146- $\delta 1$

Although the data support our hypothesis, investigation of the bound state is still necessary if the hypothesis is to be further tested. Moreover, all NS1 atoms, not just methyl-containing residues, should be compared. Arguably, this is beyond the limitations of current NMR approaches. Thus, to clearly indicate this limitation, we added the following sentences in the revised manuscript (page 8):

"However, the underlying molecular mechanism remains to be determined. In particular, future studies on the NS1-p85b complex are warranted to reveal the mechanism."

7. Similarly, does the NMR data provide insight into why mutations at the core interface residues have the least deleterious effect for Ud NS1 (Figure 2c-d)?

Similar to our response to #6, we will need a full dynamics study of the p85 β -bound state to address this question. Thus, we cautiously provide our best potential explanation as follows. We hypothesize that the p85 β -binding interface of Ud NS1 might be more rigid in the BC conformation compared to other NS1s. Thus, the mutational effects of a core interface residue would be small because other interface residues, including both the core and rim, could still interact reasonably well with p85 β . In other words, the binding interface of Ud NS1 might be better pre-organized for productive binding to p85 β than those of other NS1s.

Indeed, this hypothesis is supported by the NMR order parameters (S^2_{axis}) for methyl-containing residues in the p85 β -binding interface residues (see table above). However, the following caveats should be considered: First, only methyl-containing residues were considered in this study. Second, not all methyl-containing interface residues across NS1s were included because of the peak overlap in some NS1s. Therefore, although interesting, further follow-up studies will be needed in the future to test our hypothesis thoroughly.

Minor comments:

1. In Supplementary Fig. 1a, it will be helpful to indicate what is the % sequence identity of NS1 between each strain and the 1918 strain.

We added the % sequence identity as suggested.

2. In Supplementary Fig. 1a, it will also be helpful to indicate the locations of the core interface residues 89, 95, 98, 145, and 146.

We indicated the positions of the core interface residues in Supplementary Figure 1a.

3. In the second paragraph of the results section, is "dissociation rate constant" equivalent to "KD" in Figure 1c?

The "dissociation rate constant" corresponds to k_{off} in Figure 1c.

4. The error bars in Figures 2c-d and Supplementary Figure 4 need to be defined in the figure legends.

Throughout the manuscript, the error bars represent the standard deviation of multiple measurements or propagated uncertainty. We added the definition to the figure legends.

5. In Figure 4e, it will be useful to label the strain name next to the PDB ID to support the statement: "all available Ud NS1 structures showed only a BC-like conformer."

We added the PDB IDs of all available Ud NS1 structures.

6. Does the conformationally most heterogeneous region (marked with a red dotted circle in Figure 4c) overlap with the p85 β binding site?

Some parts of the structurally heterogeneous region correspond to the p85 β -binding site. For example, Figure A on the right shows two representative BI (magenta) and BC (cyan) conformers. When NS1 is in the BI conformation, parts of the β 1 and β 7 strands clash with p85 β (see Figure B on the right). As a result, p85 β can only bind to the BC conformer. A detailed analysis of the BI-BC conformational exchange was reported (PNAS, 2020, vol 117, 6550-6558).

Additional changes according to formatting instructions

Title: We shortened our title to “**Energy landscape reshaped by strain-specific mutations underlies epistasis in NS1 evolution of influenza A virus.**” The new title is less than 15 words.

Abstract: We shortened our abstract to conform to the word limit. The changes are highlighted in yellow in the revised manuscript.

Reviewer comments, second round review:

Reviewer #1 (Remarks to the Author):

The authors have addressed my concerns

Reviewer #2 (Remarks to the Author):

Most of my previous concerns have been addressed by the authors. However, there are still two minor concerns.

Minor comments:

1. In the sequence identity table in Supplementary Figure 1, the numbers seem incorrect. For example, it is impossible to only have 3.3% sequence identity between PR8 NS1 and 1918 NS1.
2. What do those black dots above the sequence alignment in Supplementary Figure 1 represent? Please clarify in the legend.

Reviewer #1 (Remarks to the Author):

The authors have addressed my concerns

Reviewer #2 (Remarks to the Author):

Most of my previous concerns have been addressed by the authors. However, there are still two minor concerns.

Minor comments:

1. In the sequence identity table in Supplementary Figure 1, the numbers seem incorrect. For example, it is impossible to only have 3.3% sequence identity between PR8 NS1 and 1918 NS1.

: We apologize for this mistake. We accidentally provided the sequence difference, not the identity. In the revised version, we added a matrix showing the sequence identity between all NS1s in Supplementary Figure 1. We thank the reviewer for carefully reading our manuscript.

2. What do those black dots above the sequence alignment in Supplementary Figure 1 represent? Please clarify in the legend.

: The black circles represent the positions of core binding residues. We added the following sentence in the figure legend.

“Closed circles represent the positions of core binding residues. Amino acid sequence identities between NS1s of different IAV strains are shown in a matrix format.”